# Exploring the Role of Sclerostin as a Biomarker of Cardiovascular Disease and Mortality: A Scoping Review

**DOI:** 10.3390/ijerph192315981

**Published:** 2022-11-30

**Authors:** Raquel Sanabria-de la Torre, Sheila González-Salvatierra, Cristina García-Fontana, Francisco Andújar-Vera, Beatriz García-Fontana, Manuel Muñoz-Torres, Blanca Riquelme-Gallego

**Affiliations:** 1Instituto de Investigación Biosanitaria de Granada (ibs.GRANADA), 18014 Granada, Spain; 2Department of Medicine, University of Granada, 18016 Granada, Spain; 3CIBER on Frailty and Healthy Aging (CIBERFES), Instituto de Salud Carlos III, 28029 Madrid, Spain; 4Endocrinology and Nutrition Unit, University Hospital Clínico San Cecilio, 18016 Granada, Spain; 5Department of Computer Science and Artificial Intelligence, University of Granada, 18071 Granada, Spain; 6Andalusian Research Institute in Data Science and Computational Intelligence (DaSCI Institute), 18014 Granada, Spain; 7Department of Preventive Medicine and Public Health, University of Granada, 18016 Granada, Spain

**Keywords:** biomarkers, cardiovascular disease, cardiovascular mortality, cardiovascular risk, sclerostin

## Abstract

Sclerostin is most recognized for its role in controlling bone formation; however, it is also expressed in the heart, aorta, coronary, and peripheral arteries. Human studies have associated high circulating sclerostin levels with the presence of different cardiovascular diseases (CVD), surrogate CVD markers, and a high risk of cardiovascular events in some populations. However, this is still a matter of scientific debate, as the results have been very heterogeneous among studies. In the present review, the association between serum sclerostin levels and CVD and/or cardiovascular mortality was analyzed. For this purpose, a scoping review was performed in which articles measuring serum sclerostin levels and cardiovascular risk in patients were selected. Eleven articles answered the research question; of these articles, 8/11 evaluated the association between sclerostin and CVD, of which 4/8 found a positive association, 2/8 found a negative association, and 2/8 found no association between variables. Five (5/11) of the articles included in the study evaluated cardiovascular mortality, of which 3/5 found a positive association, 1/5 found a negative association, and 1/5 found no association between variables. In conclusion, we did not find sufficient results to be able to demonstrate an association between elevated sclerostin levels and the development of CVD and/or cardiovascular mortality in the general population due to heterogeneity in the results. However, there seems to be a tendency to consider increased sclerostin levels as a risk factor for both the development of cardiovascular events and cardiovascular mortality in specific populations. Further studies in this field will help to solve some of the inconsistencies found during this scoping review and allow for the future use of sclerostin measurement as a strategy in the prevention and diagnosis of CVD and/or cardiovascular mortality.

## 1. Introduction

Cardiovascular disease (CVD) is defined as a heterogeneous group of cardiac and circulatory system disorders that generally evolve silently and chronically until they reach a late stage [1]. Mortality rates of this pathological group are expected to increase in the near future due to the relationship between its prevalence and unhealthy lifestyles [2].

At present, 17.3 million people die each year from CVD; it is considered to be the leading cause of morbidity and mortality worldwide [3,4]. In Europe, it is the cause of death of more than 4 million people annually and is more frequent among men than women. However, 80% of these deaths are preventable [5].

In this context, cardiovascular prevention is an essential factor defined as a series of coordinated actions for the multitude, group, or individuals, with the aim to reduce or minimize the impact of CVD and related disabilities in the community.

The risk factors for CVD are classified as non-modifiable (age and sex, [men over 45 years of age and women over 55 years of age at menopause], family history with positive inheritance), and modifiable risk factors, generally associated with unhealthy lifestyle habits (smoking, sedentary lifestyle, arterial hypertension, obesity, dyslipidemia, and type 2 diabetes mellitus (T2DM), among others) [6]. According to the National Institute for Health and Care Excellence (NICE) Report, a population-based approach to reducing the risk of developing CVD would lead to the prevention of other diseases, such as cancer, T2DM, and Chronic Obstructive Pulmonary Disease (COPD) [7,8], in addition to the direct cost savings from avoided cardiac events, pharmacological treatment, or medical follow-up, and indirect cost savings from reduced sick leave due to incapacity [9].

Several risk scores, such as the Framingham Risk Score (FRS) [10], the Systematic Coronary Risk Evaluation (SCORE) [11], and the atherosclerotic cardiovascular disease (asCVD) score [12], have been developed to predict the future risk of CVD and CVD mortality. Likewise, the REGICOR score is the result of the validation of the Framingham equation in the Spanish population [13,14]. These scores were designed for risk prediction in the general population, and the FRS, in particular, is widely used in clinical settings [15].

Another tool that is becoming increasingly important for the correct stratification of cardiovascular risk is the determination of biomarkers. Recently, a relationship was observed between markers classically involved in bone metabolism (osteocalcin, osteopontin, osteoclastin, vitamin D, or sclerostin) and cardiovascular risk [16,17]. Sclerostin is a soluble glycoprotein encoded by the *SOST* gene on chromosome 17q12–q21. This protein is synthesized mainly by osteocytes and, to a lesser extent, other cell types, including osteoclast precursors and renal and vascular cells [16]. Sclerostin secretion leads to a reduction in bone formation as it suppresses osteoblast activation and inhibits bone turnover [18]. This process is carried out through the Wnt/β catenin signaling pathway in bone. The activation of this pathway in bone occurs through binding of Wnt proteins to Frizzled receptors and co-receptors of the low-density lipoprotein family (LDL) 5 and 6, producing β-catenin stabilization and activating gene transcription [17]. Activation of the Wnt pathway favors osteoblastic differentiation of mesenchymal stem cells and restricts chondrogenic and adipogenic differentiation. In addition, it also leads to osteoblast maturation processes, increased osteocyte survival, and inhibition of osteoclast genesis [5]. Sclerostin levels are generally higher among men; however, these levels tend to increase in women due to the decrease of estrogen levels during menopause [16]. Most studies that have determined sclerostin levels in the healthy population have indicated a mean value of approximately 40 ± 15 pmol/L, regardless of sex and age [19,20,21,22,23].

Determination of the sclerostin level is useful in the diagnosis of different pathologies, such as Van Buchem Dam disease [18], or to study the progression of chronic kidney disease (CKD) [24].

Regarding sclerostin at the vascular system, it seems clear that this protein has an important role at the vascular level. Different studies have shown that vascular smooth muscle cells (VSMCs) can induce a phenotypic transition to osteocyte-like cells capable of expressing typical osteocytic markers, including sclerostin, in a calcifying environment [25]. Thus, sclerostin expression has been found in atherosclerotic plaques and has been linked to vascular calcification in menopausal women [26], diabetic patients [27], and in those with abnormalities in the thickness of the arterial intima-media layers [28].

At the serum level, several studies have shown an association between serum sclerostin levels and the presence of cardiovascular events [29] or cardiovascular mortality [21]. These findings show that the action of sclerostin is not only restricted to the regulation of bone formation, it is also involved in vascular integrity, constituting an important modulator of Wnt signaling in CVD and acting as a potential serum marker of cardiovascular risk [23,25,30]. 

However, there is controversy regarding the role of sclerostin in vascular tissue. Some studies have reported a pathological role of this protein, mainly in CKD patients [31,32]. Sclerostin has also been described as a mediator between physical exercise and cardiovascular risk. Recent studies have demonstrated that mechanical loading associated with high-intensity interval training (HIIT) may improve bone status and atherosclerosis parameters (i.e., carotid intima-media thickness (cIMT)) through a decrease in the serum levels of Wnt signaling inhibitors, such as Dkk-1 and sclerostin, thus decreasing cardiovascular risk [33]. The most physically active individuals have the lowest serum sclerostin levels [33,34], which could be associated with a lower cardiovascular risk in this population. However, other studies have described a substantial but transient increase in sclerostin after physical activity [33,35,36], which returns to basal levels at 1 h post-exercise [37]. It is speculated that an exercise session results in an initial catabolic response (elevated sclerostin), which is subsequently followed by an anabolic response [37].

Therefore, in the literature, we found that short-term exercise causes an increase in sclerostin levels due to a physiological anabolic response [35], and that long-term exercise can decrease sclerostin levels, which is a cardioprotective action [34].

Conversely, other authors attribute a protective role to sclerostin, which may be involved in blocking the progression of vascular calcification through its inhibitory action on the Wnt signaling pathway, as observed in animal models [38] and in human studies [39,40]. 

In this context, there is no clear physiological link between sclerostin and CVD since, despite numerous studies, no consensus has yet been reached. Therefore, the association between higher sclerostin levels and predisposing factors to the development of CVD has prompted the scientific community to study it in large observational studies, to establish whether increased levels of this protein play a protective role in the survival of patients with CVD or, on the contrary, are involved in the pathogenesis of these complications.

The importance of understanding the pathophysiological mechanisms related to this protein and its effect on the vascular physiology could lie in advancements in the early diagnosis and prevention of CVD, reducing the high morbidity and mortality of the population at a higher risk [41].

In this study, we conducted a scoping review of the existing literature on the relationship between serum sclerostin levels and the development of cardiovascular events in humans. Clinical studies determining sclerostin values and their relationship with mortality and/or cardiovascular events were selected and critically analyzed.

## 2. Materials and Methods

This study is a scoping review based on the current scientific evidence on the relationship between elevated sclerostin serum levels and cardiovascular events and/or cardiovascular mortality.

### 2.1. Research Question

The design of a scoping review with systematic methodology has been followed. This type of review is a form of evidence synthesis that follows the process of a systematic review, in which some steps are simplified or omitted to produce information in a short period of time on the more specific elements related to the research objective [42].

To report the main findings of this review, we followed the verification guidelines established in the PRISMA (Preferred Reporting Items for Systematic Reviews) guide, in its 2020 update [43], adapting some items due to the chosen review design [42,43,44]. 

The main research question was as follows: Are elevated sclerostin values a risk exposure factor for CVD?

The PICO question was formulated for the development of the research question, focusing the search for original articles to demonstrate the main issue.

P = articles studying sclerostin values and their relationship with cardiovascular mortality and/or cardiovascular events;I = elevated baseline sclerostin values as a factor of exposure;C = normal sclerostin values *;O = severe cardiovascular events: cardiovascular death, acute myocardial infarction, stroke, heart attack, myocardial infarction, stroke, angina pectoris;S = cohort, case-control, and cross-sectional studies.

* Mean serum sclerostin levels in men and women were 40 ± 15 pmol/L, which was independent of age and sex [19,20,21,22,23].

### 2.2. Databases Consulted

The main databases consulted for the detailed search of the scientific evidence were PUBMED, SCOPUS, and Web of Science, where the following keywords were used: Sclerostin and Cardiovascular Disease. In PUBMED, the thesaurus developed by the National Library of Medicine (NLM), called Medical Subject Headings (MeSH), was used. In the SCOPUS and Web of Science databases, the thesaurus of Descriptors in Health Sciences (DeCS) was chosen. The search equation was formulated depending on the focused database and using the Boolean operator AND, resulting in the following search engine: (Sclerostin) AND “Cardiovascular Diseases” [Mesh] in PUBMED and “Sclerostin” AND “Cardiovascular diseases” in SCOPUS and Web of Science. Once the search string was obtained, the keywords were consulted separately in the different databases. The results obtained are shown in Table 1.

### 2.3. Inclusion and Exclusion Criteria

The inclusion criteria were as follows: (1) freely published articles or those that can be accessed through the library of the University of Granada, (2) articles where the population studied were adults, (3) articles relating serum sclerostin levels to death from cardiovascular causes or severe cardiovascular events, and (4) written in English or Spanish language. The exclusion criteria were as follows: (1) articles performed in animal models, (2) articles in which a pediatric population was studied, (3) articles where serum sclerostin levels were not measured, and (4) articles that did not show a severe cardiovascular event or death from cardiovascular cause.

### 2.4. Selection of Studies

To carry out the study selection, a two-step review was applied. The titles and abstracts were screened independently by two authors (RST and BRG), and all publications reporting the measurement of sclerostin levels and CV risk and/or CV mortality and/or CV events were included. Secondly, the entire publication was reviewed to ensure that it fulfilled the rest of the inclusion criteria. Disagreements about inclusion or exclusion of articles were discussed until a consensus was reached.

### 2.5. Data Extraction

The variables assessed were author and date, country, age, number of participants, study design, follow-up, percentage of diabetes patients, and percentage of dialyzed patients. Sclerostin levels were expressed in pmol/L in four articles, in pg/mL in five articles, in ng/mL in one article, and in μg/L in another. Due to the diversity of units of measurement, the results were unified in ng/mL. To analyze the association between sclerostin levels and cardiovascular events, data related to CVD and cardiovascular mortality of the patients included in the selected studies were extracted. 

### 2.6. Risk of Bias Assesment

The evaluation of the methodological quality of the selected articles was carried out with the Critical Appraisal Skills Programme Español (CASPe) tool. CASPe is the most used tool for quality appraisal in health-related qualitative evidence syntheses [45]. The CASPe checklist for assessing the quality of observational studies has 11 items designed to assist in the response to three questions to consider when assessing a cohort, case-control, or cross-sectional study: Section A: Are the results of the study valid; Section B: What are the results; Section C: Will the results help at the local level? The first two questions are screening questions and can be answered quickly. If the answer to both is “yes”, it is worth continuing with the remaining questions. You are asked to record a “yes”, “no” or “can’t say” to most of the questions.

## 3. Results

### 3.1. Selection and Description of Studies

On 21 December 2021, a bibliographic search was performed using the databases mentioned above (Section 2.2). The initial sample of data consisted of a total of 298 articles. A screening of the bibliography was carried out by means of a previous reading of the title and abstract, obtaining those articles that focused on the line of research to be dealt with.

After the initial selection of the articles, we proceeded to read them completely, discarding those whose characteristics did not meet the inclusion/exclusion criteria previously established. Finally, duplicate studies were eliminated. The final sample consisted of 11 articles with a total population of 2786 patients (Figure 1). The main patient information from the selected articles is summarized in Table 2.

Nine of the studies were longitudinal [21,40,46,47,48,49,50,51,52], and two were case-control [53,54]. The duration of follow-up ranged from 7 months [54] to 15.5 years [51]. The percentage of male patients ranged from 43.87% [40] to 68% [49] of the study population. The mean age of the patients ranged from a minimum of 52.5 years [47] to a maximum of 83 years [21]. The presence of diabetes was measured nine out of 11 studies [51,53], with the prevalence of diabetes ranging from 7.28% [52] to 57.7% [21]. The percentage of patients undergoing dialysis was measured in six out of 11 articles, in four of which all patients were undergoing dialysis [46,47,50,51] and two of which 54.6% [48] and 62.4% [49] were undergoing dialysis.

**Table 2 ijerph-19-15981-t002:** Characteristics of the studies included.

Author and Date	Country	N	Male Patients (%)	Average Age (Years)	Diabetes (%)	Dialysis (%)	Follow-Up (Months)	Ref.
Zou Y. et al., 2020	China	165	55.8	56.5	29.1	100	24.9	[50]
Novo-Rodríguez C. et al., 2018	Spain	130	51.5	58.8	57.69	N/C	82	[21]
Kalousová M. et al., 2019	Czech Republic	106	60.38	61	40.6	100	26.5	[46]
Gong L. et al., 2018	China	98	49	52.5	21.4	100	72	[47]
Wang X-R. et al., 2017	China	161	49.38	58.3	20.5	54.66	23	[48]
Jørgensen HS. et al., 2018	Denmark	157	68	54	26	62.42	36	[49]
He W. et al., 2020	China	310	43.87	76	35.48	N/C	36	[40]
Klingenschmid G. et al., 2020	Italy	706	47.6	66.3	N/C	N/C	187.2	[51]
Drechsler C. et al., 2015	Netherlands	673	57	63	7.28	100	48	[52]
He XW. et al., 2016	China	186	61.3	71.7	20.43	N/C	7	[54]
Mathold K. et al., 2018	Sweden	94	50	83	N/C	N/C	N/C	[53]

### 3.2. Methodological Quality

The statistical analyses used in each study are summarized in Table 3. The co-variates included in the analyses to adjust the dataset were also collected. Furthermore, after evaluating the methodological quality of the selected studies according to the CASPe tool, a moderate score (8–9/11) was found in four of the articles [46,48,53,54] and a high methodological score (10–11/11) in seven of the remaining articles [21,40,47,49,50,51,52] (see Table 4 and Table 5).

### 3.3. Association between Sclerostin Levels and Cardiovascular Events

Eight of the articles studied the relationship between sclerostin levels and the development of CVD. Four of the studies consisted of a total of 543 patients, including two cohorts Zou Y. et al., 2020 [50] (in the peritoneal dialysis population), Gong L. et al., 2018 [47], and two case-control studies by He X.W. et al., 2016 [54] and Mathold K. et al., 2018 [53], showed a positive significant relationship between sclerostin levels and the development of CVD (Table 4). On the other hand, two of the studies showed a negative association between sclerostin levels and the development of CVD (Table 4) (Wang X-R. et al., 2017 [48], He W. et al., 2020 [40]); the other two found no association between these variables (Table 4) (Jørgensen H.S. et al., 2018 [49] and Klingenschmid G. et al., 2020 [51]). Most studies analyzed the relationship between sclerostin levels as a continuous variable and the occurrence of cardiovascular events with the exception of the studies by Jørgensen H.S. et al. 2018 [49] and He X.W. et al., 2016 [54], which used average sclerostin values.

### 3.4. Association between Sclerostin Levels and Cardiovascular Mortality

Five articles prospectively studied the relationship between sclerostin levels and cardiovascular mortality [21,40,46,47,52]. In three longitudinal studies with a total of 334 patients, a significant positive relationship was found between both variables (Table 5). On the other hand, in the study by Drechsler C. et al., 2015 [52], an inverse relationship between sclerostin levels and cardiovascular mortality in patients with CKD undergoing dialysis was found (Table 5). Finally, in the study of He W. et al., 2020 [40], no relationship was found between sclerostin levels and cardiovascular mortality (Table 5). 

**Table 3 ijerph-19-15981-t003:** Statistical analysis and co-variables used in each study.

Author and Date	Type of Analysis	Binary CVD (Yes/No)	Co-Variates Included in the Model	Ref.
Drechsler C. et al., 2015	Competing risk analysis	No	Age, sex, residual GFR, blood pressure, levels of serum albumin, haemoglobin, calcium, PTH, AP	[52]
Wang X.R. et al., 2017	Univariate and Multivariate Cox regression	Yes	Diabetes, smoking habit, dialysis status	[48]
Mathold K. et al., 2018	One-way ANOVA and ANCOVA	Yes	Age, sex, waist circumference, eGFR	[53]
Jørgensen HS. et al., 2018	Univariate and Multivariate Cox regression	Yes	Age, sex, BMI, dialysis therapy	[49]
Gong L. et al., 2018	Univariate and Multivariate Cox regression	Yes	Age, sex, diabetes mellitus, LVEF, NT-proBNP, Pi, PTH, OPG	[47]
Novo-Rodríguez C. et al., 2018	Competing risk analysis	Yes	Age, diabetes, sex, prevalent CVD, pIMT, tobacco use, hypertension, eGFR	[21]
Kalousová M. et al., 2019	Linear regression model and real survival curves	Yes	Age	[46]
He W. et al., 2020	Univariate and Multivariate Cox regression	Yes	Cardiovascular risk factors (TG, TC, LDL-c, HDL-c, hs-CRP, HbA1c, creatine), osteoporosis, CAD severity, angiographic characteristics, use of medication	[40]
Klingenschmid G. et al., 2020	Linear regression models	Yes	Age, sex	[51]
Zou Y. et al., 2020	Univariate and multivariate Cox regression analysis	Yes	Age, sex, Framingham cardiovascular risk factors (smoking, diabetes status, BP, HDL-c), factors associated with mortality in patients with CKD (CRP, phosphate, BMI, and albumin), and confounding factors associated with mortality, new onset CVEs	[50]
He XW. et al., 2016	Multiple logistic regression analysis	Yes	BMI, hypertension, diabetes mellitus, dyslipidaemia, smoking, drinking, homocysteine, hs-CRP	[54]

GFR: glomerular filtration rate; BMI: body mass index; hs-CRP: hypersensitive C-reactive protein; CVEs: cardiovascular events; CKD: Chronic kidney disease; CRP: C-Reactive Protein; BP: blood pressure; PTH: parathyroid hormone; AP: alkaline phosphatase; TG: triglyceride; TC: total cholesterol; LDL-c: LDL-cholesterol; HDL-c: HDL-cholesterol; HbA1c: glycated hemoglobin; CAD: coronary artery disease.

**Table 4 ijerph-19-15981-t004:** Association between sclerostin levels and CVD.

Author and Date	Country	N	Study Type	Population	Scl Level (ng/mL)	Cox Proportional HR Analysis CV Events	Scl-CVD	CASPe
Zou Y. et al., 2020 [50]	China	165	Longitudinal	HD; PD	0.2509	HD patients (HR = 1.164, *p* = 0.509) PD patients (HR = 3.819, *p* = 0.011)	+	10/11
Gong L. et al., 2018 [47]	China	98	Longitudinal	PD	1.945	UA (HR = 2.456, *p* = 0.013); MA (HR = 2.475, *p* = 0.026)	+	10/11
Wang X-R. et al., 2017 [48]	China	161	Longitudinal	Stage 3,4 and 5 CKD patients	0.898	MA (HR = 0.294, *p* = 0.001)	−	9/11
Jørgensen H.S. et al., 2018 [49]	Denmark	157	Longitudinal	Candidates for renal transplantation due to CKD	0.259	MA (HR = 0.99, *p* = 0.88)	NS	10/11
He W. et al., 2020 [40]	China	310	Longitudinal	Geriatric patients undergoing PCI	0.179	MA (HR = 0.456, *p* = 0.013)	−	11/11
Klingenschmid G. et al., 2020 [51]	Italy	706	Longitudinal	Bruneck area population	1.07	UA (HR = 0.95, *p* = 0.507); MA (HR = 0.92, *p* = 0.507)	NS	11/11
He X.W. et al., 2016 [54]	China	186	Case-control	LAA and SAO incident stroke patients	LAA: 0.15; SAO: 0.15	ROC Curve Analysis AUC = 0.773, *p* < 0.001	+	9/11
Mathold K. et al., 2018 [53]	Sweden	94	Case-control	Geriatric patients with N-CEIS vs. healthy controls.	Cases: 3; Controls: 1.9	ANOVA 3.0 (1.0–7.3) vs. 2.0 (0.2–6.5), *p* < 0.001	+	8/11

Scl: Sclerostin; CV: Cardiovascular; CVD: Cardiovascular disease; CKD: Chronic Kidney Disease; PCI: Percutaneous Coronary Intervention; LAA: Large Artery Atherosclerosis; SAO: Small-Artery Occlusion; HD: Hemodialysis; PD: Peritoneal Dialysis; HR: Hazard Regression; ROC: Receiver Operating Characteristic; AUC: Area Under the Curve; N-CEIS: Non-Cardioembolic Ischemic Stroke; UA: Univariate Analysis; MA: Multivariate Analysis. CASPe: The Critical Appraisal Skills Programme Español tool; NS: non-significant.

**Table 5 ijerph-19-15981-t005:** Association between sclerostin levels and cardiovascular mortality.

Author and Date	Country	N	Study Type	Population	Scl Level (ng/mL)	Cox Proportional HR Analysis Mortality	Scl-CV Mortality	CASPe
Novo-Rodríguez C. et al., 2018 [51]	Spain	130	Longitudinal	DP	1.317	HR = 1.318, *p* = 0.004	+	10/11
Kalousová M. et al., 2019 [46]	Czech Republic	106	Longitudinal	HD	1.90	HR = 2.557, *p* = 0.04	+	9/11
Gong L. et al., 2018 [47]	China	98	Longitudinal	PD	1.945	UA (HR = 4.362, *p* = 0.008); MA (HR = 3.484, *p* = 0.029)	+	10/11
He W. et al., 2020 [40]	China	310	Longitudinal	Geriatric patients undergoing PCI	0.179	N/C	NS	11/11
Drechsler C. et al., 2015 [52]	Netherlands	673	Longitudinal	CKD patients with HD and PD	1.37	Short term (HR = 0.30 (0.14–0.61); Long term (HR = 0.60 (0.37–0.97)	−	10/11

Scl: Sclerostin; CV: Cardiovascular; DP: diabetic patients; HD: hemodialysis; PD: peritoneal dialysis; PCI: percutaneous coronary intervention; CKD: chronic kidney disease; UA: Univariate Analysis; MA: Multivariate Analysis. CASPe: The Critical Appraisal Skills Programme Español tool; NS: non-significant.

## 4. Discussion

The present scoping review evaluated the association between elevated serum sclerostin levels and the development of cardiovascular events and cardiovascular mortality. Eleven articles in which sclerostin levels were evaluated and related to the development of CVD and cardiovascular mortality were analyzed.

The relationship between serum sclerostin levels and CVD has been addressed by eight studies, six of which were longitudinal and two of which were case-control studies. The results of the case-control studies showed a direct relationship between circulating levels of sclerostin and the presence of cardiovascular events and cardiovascular risk. The longitudinal studies showed an equal proportion of positive results in terms of association between elevated sclerostin levels and cardiovascular risk (n = 2), negative results (n = 2), and results with no association between the study variables (n = 2). Regarding the relationship between circulating sclerostin and cardiovascular mortality, this aspect was evaluated in five longitudinal studies; three of them showed an independent association between elevated sclerostin levels and cardiovascular mortality, while one showed the opposite result, and one showed no association between the study variables. Although the results of the cross-sectional studies do not imply causality, the absence of significant results in some of the longitudinal studies does not imply the inexistence of a relationship, but rather suggests that there is insufficient evidence to validate the proposed hypothesis. There does seems to be a tendency towards increased sclerostin levels as a risk factor, both for development of cardiovascular events and cardiovascular mortality, since more than half of the analyzed studies showed a significant association between the study variables.

The heterogeneity of the results obtained is probably due to the comorbidities associated with the studied patients, which does not allow us to draw solid conclusions for the general population. However, the observed trend of increased sclerostin levels as a cardiovascular risk factor seems to be more pronounced in specific populations. Thus, the literature suggests a positive association between high sclerostin serum levels and the development of CVD in patients with incident stroke. Likewise, serum sclerostin levels of patients undergoing peritoneal dialysis and hemodialysis were positively associated with the development of CVD but not with cardiovascular mortality. On the other hand, studies analyzing a Brunek area population-based cohort did not find a significant relationship with our initial hypothesis, as well as those studying patients with CKD and patients undergoing percutaneous coronary intervention. 

The pathophysiological process that determines both the development of CVD and cardiovascular mortality in relation to elevated sclerostin levels is the most uncertain aspect of the proposed hypothesis. Researchers, such as Brandenburg V.M. et al., 2013 [55], suggest that the cause of the cardiovascular events is increased cardiovascular calcification due to sclerostin elevation; this was also pointed out in six of the 11 reviewed articles including patients with renal pathology. Pelletier et al., 2013 [56] showed an inverse relationship between the serum concentration of sclerostin and the glomerular filtration rate in these patients. In addition, the levels of sclerostin are modified due to the presence of dialysis as a renal therapeutic measure. Kalousová et al., 2019 [46] found that mean sclerostin levels in hemodialysis patients were almost three times higher than in patients with adequate renal function. Some authors have debated the cause of this process, i.e., whether it is due to reduced renal clearance, increased skeletal production, or even extra-skeletal production is still a subject of debate [24]. The hypothesis put forward by many authors is the mediation of high sclerostin values as a fundamental cause of bone calcification in the progression of CKD [57]. In fact, evidence has suggested that increased sclerostin may have a positive association with aortic calcification, abnormal intima-media thickness, high arterial stiffness, and carotid plaques in T2DM patients [29] and in healthy subjects [58]. The upregulation of sclerostin in calcified tissues led researchers and clinicians to come up with a hypothesis that sclerostin may be related to the pathogenic mechanism of vascular calcification in renal disease patients. 

However, the role of sclerostin at the vascular level has not yet been fully determined and other studies show opposite results. Although recent reviews point out the association between increased sclerostin levels and large carotid intima media thickness, severe vascular calcification, and high arterial stiffness, as well as with an increased prevalence of atherosclerotic plaques and the development of other cardiovascular events [58,59], the role of sclerostin as an inhibitor of mineralization could suggest that the increase in sclerostin could be a reflection of a compensatory mechanism to slow down vascular calcification. This fact puts the role of sclerostin at the vascular level as a focus of current research. In this context, a recent review reported a protective role of sclerostin against abdominal aortic aneurysms and atherosclerosis formation in preclinical models [58]. Human genetic studies reported that low arterial sclerostin expression was associated with a high risk of cardiovascular events [58]. However, it has not been possible to obtain a clear conclusion due to the diversity of the results obtained in several studies [21,40,46,47,48,49,50,51,52,53,54]. 

In this context, the anti-sclerostin monoclonal antibody Romosozumab is currently under investigation. This antibody promotes bone formation by inhibiting sclerostin activity. The results have shown an increase in bone mineralization and avoidance of fractures in those at high risk, especially in the postmenopausal population. However, serious cardiovascular adverse effects were reported during the study with Romosozumab, which agrees with the results obtained by Gay A. and Towler D.A. in 2017 [60]. However, other meta-analyses of randomized controlled clinical trials suggested that administration of Romosozumab did not significantly increase the risk of major adverse cardiovascular events [61,62]. Thus, there is a high level of controversy regarding the cardiovascular safety of this drug. In fact, the data available to date supports restricting the prescription recommendations detailed in the drug’s data sheet, stating that patients at high risk of cardiovascular disease and stroke should not be considered for treatment with Romosozumab.

Due to the heterogeneity in the relationship between sclerostin and cardiovascular risk shown in numerous studies, perhaps there is a spurious relationship between these variables. There could be a third confounding factor that affects the two initial variables. It is therefore possible that there is not a simple unilateral relationship between sclerostin and cardiovascular risk, since there could be other underlying factors that are not currently being evaluated, as occurs in other processes detailed in scientific literature [63].

Based on our results and the results of other studies, we encourage further research to obtain more robust results to elucidate the involvement of this protein in vascular tissue and its relationship with CV risk and mortality, considering its potential clinical importance in the prevention of CVD, survival rate, and the quality of life of patients. 

Deepening the studies in this field will help to solve some of the inconsistencies found during this literature review and allow for the future use of sclerostin measurement as a strategy in the diagnosis and prevention of CVD and/or cardiovascular mortality. 

### 4.1. Limitations

Due to the diversity of procedures used to analyze the sclerostin levels, certain difficulties arose when unifying the mean values, and it is advisable to establish uniform criteria that facilitate understanding for health care professionals. The differences in terms of sample size, type of population, underlying pathologies at an average age, medication, comorbidities, and duration of follow-up could contribute to the appearance of biases, which may make it difficult to obtain comparable results between groups.

On the other hand, the present review, despite presenting a systematic approach, did not comply with all the items recommended by the PRISMA guidelines; therefore, it may have certain methodological limitations.

Furthermore, a possible spurious relationship between sclerostin levels and the incidence of cardiovascular events cannot be ruled out. The main limitation of this study was the inconsistency of the results obtained, since we have found articles whose results agreed with this hypothesis, articles that deny it, and articles that did not find a statistical association between the outcome variables.

### 4.2. Perspectives

The search process through the different scientific databases, covering a wide spectrum of the current literature, as well as the methodological evaluation of the selected articles through CASPe, allowed us to objectively analyze the results obtained in these studies and weigh them for their methodological quality, which added robustness to the results shown.

In this context, it is hoped that the present literature review will be useful as an incentive for researchers to carry out further in-depth analysis of the potential of sclerostin. 

## 5. Conclusions

Due to the heterogeneity of the results obtained in the present review, we cannot robustly demonstrate a relationship between elevated sclerostin serum levels and the development of CVD and/or cardiovascular mortality in the general population; however, there does appear to be a positive trend in specific populations. Because of the clinical importance of this hypothesis, we encourage further research along these lines.

## Figures and Tables

**Figure 1 ijerph-19-15981-f001:**
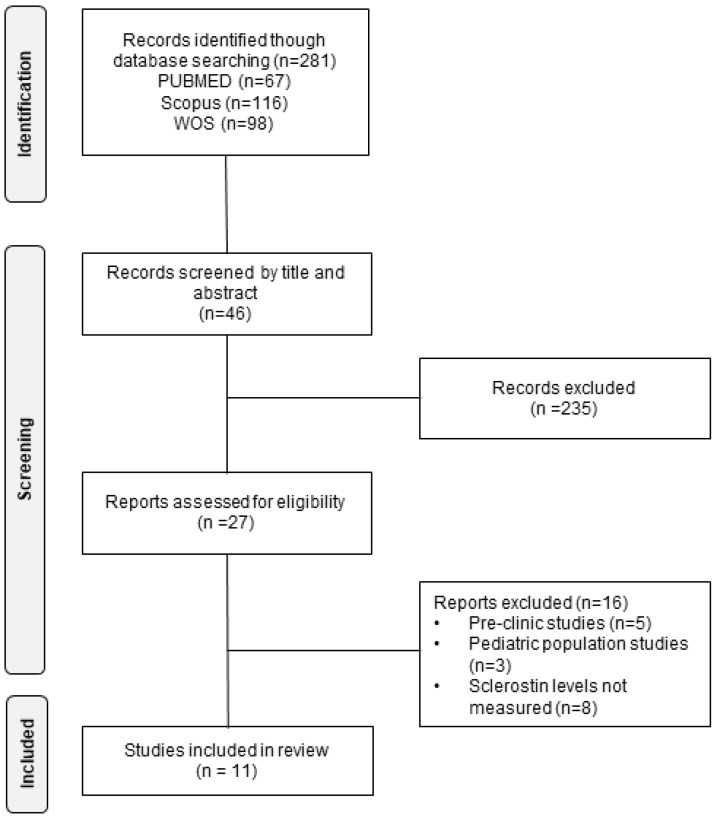
Flow chart of the literature search performed for the selection of the studies included in the analysis.

**Table 1 ijerph-19-15981-t001:** Primary search results.

	Pubmed	Scopus	Web of Science
Sclerostin	1165	3088	3078
Cardiovascular disease	406,620	654,772	390,240
Sclerostin AND Cardiovascular disease	67	116	98

Number of articles obtained by searching the Pubmed, Scopus, and Web of Science databases by entering the selected keywords both individually and combined.

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
