# Peer review of "Exploring the Role of Sclerostin as a Biomarker of Cardiovascular Disease and Mortality: A Scoping Review"

_ijerph, 2022, doi:10.3390/ijerph192315981_

Round 1

Reviewer 1 Report

The author didn't summarize any role about sclerostin in cardiovascular disease, which is lack of novelty and has no help for understanding this gene function in vascular area. This type of review is not suitable for this journal.

Author Response

Remark 1: The author didn't summarize any role about sclerostin in cardiovascular disease, which is lack of novelty and has no help for understanding this gene function in vascular area. This type of review is not suitable for this journal.

Thank you very much for this comment.

According to your comment, a more in-depth discussion of the pathophysiological role of sclerostin on cardiovascular events has been included in the introduction section as follows:

Regarding sclerostin at the vascular system, it seems clear that this protein has an important role at the vascular level. Different studies have shown that vascular smooth muscle cells (VSMCs) can induce a phenotypic transition to osteocyte-like cells capable of expressing typical osteocytic markers, including sclerostin, in a calcifying environment [25]. Thus, sclerostin expression has been found in atherosclerotic plaques and it has been linked to the vascular calcification in menopausal women [26], in diabetic patients [27] and with abnormalities in the thickness of the arterial intima-media layers [28].

At serum level, several studies have shown an association between serum sclerostin levels and the presence of cardiovascular events [29] or cardiovascular mortality [21]. These findings show that the action of sclerostin is not only restricted to the regulation of bone formation, but also appears to be involved in vascular integrity, constituting an important modulator of Wnt signaling in CVD, acting as a potential serum marker of cardiovascular risk [30–32].

However, there is controversy about the role of sclerostin at vascular tissue. Some studies have reported a pathological role of this protein mainly in CKD patients [33,34]. In this way, sclerostin has been too pointed as a mediator between physical exercise and cardiovascular risk. Recent studies have demonstrated that mechanical loading associated with high-intensity interval training (HIIT) may improve bone status and atherosclerosis parameters (i.e., carotid intima-media thickness (cIMT)) by a decrease of Wnt signaling inhibitors serum levels such as Dkk-1 and sclerostin, thus decreasing cardiovascular risk [35]. So, the most physically active individuals have the lowest serum sclerostin levels [35,36], which could be associated with a lower cardiovascular risk in this population. However, other studies have described a substantial but transient increase in sclerostin after physical activity [35,37,38], which returns to basal levels after 1 hour post-exercise [39]. It is speculated that the exercise session resulted in an initial catabolic response (elevated sclerostin), which was subsequently followed by an anabolic response [39].

Therefore, in the literature we find that short-term exercise causes an increase in sclerostin levels due to a physiological anabolic response [37]; and long-term exercise is able to decrease sclerostin levels favoring a cardioprotective action [36].

Conversely, other authors attribute a protective role to sclerostin, which may be involved in blocking the progression of vascular calcification through its inhibitory action on the Wnt signaling pathway as observed in animal models [40] as well as in human studies [41,42].

In this context, there is no clear physiological link of sclerostin in CVD, since, despite numerous studies, no consensus has yet been reached.”

Reviewer 2 Report

The manuscript titled " Exploring the role of sclerostin as a biomarker of cardiovascular disease and mortality: A scoping review." is a well thought out study with clinical relevance. The authors did literature review of the already published original articles. They found a positive relationship in the levels of sclerostin with CVD risk. However there is need for elaborate studies to confirm this finding. 

The article is well-written. A few suggestions to improve it 

1) Do the authors have any comments on the normal levels of sclerostin in heathy individuals? Are there any studies or data in it? If so, please include it in the discussion.

2)  In lines 44-46, please include appropriate references

3) In line 64, it is written "fatal CVD". Please define. Also the sentence seem bit confusing.

Author Response

The manuscript titled " Exploring the role of sclerostin as a biomarker of cardiovascular disease and mortality: A scoping review." is a well thought out study with clinical relevance. The authors did literature review of the already published original articles. They found a positive relationship in the levels of sclerostin with CVD risk. However there is need for elaborate studies to confirm this finding. 

The article is well-written. A few suggestions to improve it 

First, we want to thank for your effort in reviewing our manuscript and for your constructive comments, which have undoubtedly contributed to improve the quality of our manuscript. Please, find below the responses to your kind suggestions:

Remark 1: Do the authors have any comments on the normal levels of sclerostin in heathy individuals? Are there any studies or data in it? If so, please include it in the discussion.

According to your comment, several studies that determine serum sclerotin values in healthy subjects both in men and women have been reviewed. One of them measured serum sclerostin levels in a population-based sample of 318 men and 362 women, with age range from 21 to 97 years. Sclerostin levels (mean±SEM) were significantly higher in men than women (33.3±1.0 pmol/L versus 23.7±0.6 pmol/L, respectively) (19).  Other studies confirm these mean values in healthy subjects, reporting circulating sclerostin concentrations of 40.5 (34.8-46.2) pmol/L (20), 41.99±16.60 pmol/L (21), 52.0 (49.0-54.9) pmol/L (22); for both sexes and an age range from 40 to 60 years. Likewise, other study reported a mean circulating sclerostin of 34.23 ± 17.29 pmol/L in healthy women aged 60.8 ± 6.3 years (23). Considering these data, we conclude that the mean serum sclerostin value in healthy subjects irrespective of sex and age is about 40 ±15 pmol/L.

Thus, this value has been updated in the methods section and the following paragraph has been included in the introduction section as follows:

“Most studies that have determined sclerostin levels in the healthy population have indicated a mean value of approximately 40 ±15 pmol/L regardless of sex and age [19–23]”

References:

Mödder, U.I.; Hoey, K.A.; Amin, S.; McCready, L.K.; Achenbach, S.J.; Riggs, B.L.; Melton, L.J.; Khosla, S. Relation of Age, Gender, and Bone Mass to Circulating Sclerostin Levels in Women and Men. Journal of Bone and Mineral Research 2011, 26, 373–379, doi:10.1002/JBMR.217.

Kanbay, M.; Siriopol, D.; Saglam, M.; Kurt, Y.G.; Gok, M.; Cetinkaya, H.; Karaman, M.; Unal, H.U.; Oguz, Y.; Sari, S.; et al. Serum Sclerostin and Adverse Outcomes in Nondialyzed Chronic Kidney Disease Patients. J Clin Endocrinol Metab 2014, 99, E1854–E1861, doi:10.1210/JC.2014-2042.

Novo-Rodríguez, C.; García-Fontana, B.; de Dios Luna-Del Castillo, J.; Andújar-Vera, F.; Avila-Rubio, V.; García-Fontana, C.; Morales-Santana, S.; Rozas-Moreno, P.; Muñoz-Torres, M. Circulating Levels of Sclerostin Are Associated with Cardiovascular Mortality. PLoS One 2018, 13, doi:10.1371/JOURNAL.PONE.0199504.

Fassio, A.; Idolazzi, L.; Viapiana, O.; Benini, C.; Vantaggiato, E.; Bertoldo, F.; Rossini, M.; Gatti, D. In Psoriatic Arthritis Dkk-1 and PTH Are Lower than in Rheumatoid Arthritis and Healthy Controls. Clin Rheumatol 2017, 36, 2377–2381, doi:10.1007/S10067-017-3734-2.

Behets, G.J.; Viaene, L.; Meijers, B.; Blocki, F.; Brandenburg, V.M.; Verhulst, A.; D’Haese, P.C.; Evenepoel, P. Circulating Levels of Sclerostin but Not DKK1 Associate with Laboratory Parameters of CKD-MBD. PLoS One 2017, 12, doi:10.1371/JOURNAL.PONE.0176411.

Remark 2:  In lines 44-46, please include appropriate references

According to your comment, the following references have been included:

  • Feigin, V.L.; Roth, G.A.; Naghavi, M.; Parmar, P.; Krishnamurthi, R.; Chugh, S.; Mensah, G.A.; Norrving, B.; Shiue, I.; Ng, M.; et al. Global Burden of Stroke and Risk Factors in 188 Countries, during 1990–2013: A Systematic Analysis for the Global Burden of Disease Study 2013. Lancet Neurol. 2016, 15, 913–924, doi:10.1016/S1474-4422(16)30073-4.
  • Masters, R.K.; Powers, D.A.; Link, B.G. Obesity and US Mortality Risk Over the Adult Life Course. J. Epidemiol. 2013, 177, 431–442, doi:10.1093/aje/kws325.

Remark 3: In line 64, it is written "fatal CVD". Please define. Also the sentence seem bit confusing.

We fully agree with this comment. Accordingly, the sentence has been changed in the following way for a better understanding:

“Several risk scores, such as the Framingham Risk Score (FRS) [10], the Systematic Coronary Risk Evaluation (SCORE) [11] and the atherosclerotic cardiovascular disease (asCVD) score [12], have been developed to predict future risk for CVD, CVD mortality, as well as for asCVD”

Reviewer 3 Report

Introduction:

“SOST gene” I believe that names of genes should be provided in italic

In my humble opinion, the pathological mechanism by which Sclerostin level might play in risk of CVD disease should be better described. What is the link if sclerostin is a product of a different organ than heart? Otherwise, maybe it is a just spurious relationship: resistance physical exercise influence on the level of  Sclerostin and in the same time physical exercise decrease CVD risk? However, there is no clear physiological link of Sclerostin CVD?

Results

I think that Figure 1 could be modified in a way to look more like that proposed from CONSORT or PRISMA guidelines regarding systematic review. However, in the current form is also ok, if the Academic Editor do not see any problem with this, I would stay with the current form.

I have three technical comments:

1. Providing p values from particular studies I rather not meaningful, as they should not be compared between different studies.

2. Regarding models used in particular studies: how much of them assess the sole relationship between Sclerostin and CVD as a binary value (yes/no)? How many of those models used continuous variables for CVD risk indicator? How many of those models include co-variates? (DB2, untreated hypertension etc)? If so, what are the specific covariates included?

3. This one is about the ritual that was established in statistical analysis in medicine: If, for instance, someone applied a linear correlation model and the p-value is 0.05 or more, then it cannot be concluded that this is an evidence for no relationship. The conclusion is that there is no enough evidence to reject the null hypothesis, which states that there is no relationship. Please be aware about wording in Your paper.

Table 2: Please add (5) to male patients)

Discussion:

I would definitely add a part were You would discuss the results obtained from a  particular study in relation to the experimental model (cross sectional vs longitudinal). Do the conclusions differ between cross sectional vs longitudinal?

I would also add more information on mechanism and presumable spurious relationship, that I have mentioned about already. Please refer to a paper “Adipokines Level and Cognitive Function—Disturbance in Homeostasis in Older People with Poorly Managed Hypertension: A Pilot Study” https://doi.org/10.3390/ijerph19116467: where a potentially spurious relationship between adipokines with functional performance of older people is discussed.

Author Response

Thank you very much for your suggestions. Please, find enclosed a word file with authors comments.

Round 2

Reviewer 1 Report

The added discussion about the role of sclerostin in the vascular system doesn't change the novelty of this paper in the field.

Author Response

Dear reviewer,

We appreciate your comments and feedback. In the revised version of our manuscript we have included new references suggested by the editor and discussed the role of sclerostin in vascular pathophysiology.

Our review is in agreement with some of studies in this line, emphasizing the need to further study the role of sclerostin in cardiovascular pathophysiology taking into account that sclerostin blocking drugs are currently being used in clinical practice as anti-osteoporotic treatment that could have repercussions at the vascular level.

Finally, we would like to note that despite the numerous studies aimed at elucidating the function of sclerostin in the vascular system, to date it is still unknown due to the heterogeneity of the results. Although it is not possible to draw a robust conclusion on the relationship between sclerostin and cardiovascular disease, this review, which includes articles of high methodological quality, shows a trend towards a positive relationship between elevated serum sclerostin levels and higher cardiovascular risk, since more than half of the analyzed studies show a significant association between the study variables.

Regarding possible errors in the English language, our manuscript has been sent to MDPI's English editing service and we are awaiting receipt of the manuscript with corrections. We hope that after these corrections, the understanding of our manuscript will be easier for the readers.